# Spectroscopic Benchmarks by Machine Learning as Discriminant Analysis for Unconventional Italian Pictorialism Photography

**DOI:** 10.3390/polym16131850

**Published:** 2024-06-28

**Authors:** Claudia Scatigno, Lorenzo Teodonio, Eugenia Di Rocco, Giulia Festa

**Affiliations:** 1CREF–Museo Storico della Fisica e Centro Studi e Ricerche Enrico Fermi, Via Panisperna 89a c/o P.za del Viminale 1, 00184 Roma, Italy; lorenzo.teodonio@gmail.com; 2Freelance Restorer CRAF–Centro di Ricerca e Archiviazione della Fotografia, Piazza Castello, 33097 Spilimbergo, Italy; eugenia.dirocco@gmail.com

**Keywords:** Italian photographs, X-ray fluorescence and infrared analysis spectroscopy, kernel density, machine learning data analysis, raw data analysis, Italian pictorialism

## Abstract

Up to the 1930s, the Italian pictorialism movement dominated photography, and many handcrafted procedures started appearing. Each operator had his own working method and his own secrets to create special effects that moved away from the standard processes. Here, a methodology that combines X-ray fluorescence and infrared analysis spectroscopy with unsupervised learning techniques was developed on an unconventional Italian photographic print collection (the Piero Vanni Collection, 1889–1939) to unveil the artistic technique by the extraction of spectroscopic benchmarks. The methodology allowed the distinction of hidden elements, such as iodine and manganese in silver halide printing, or highlighted slight differences in the same printing technique and unveiled the stylistic practice. Spectroscopic benchmarks were extracted to identify the elemental and molecular fingerprint layers, as the oil-based prints were obscured by the proteinaceous binder. It was identified that the pigments used were silicates or iron oxide introduced into the solution or that they retraced the practice of reusing materials to produce completely different printing techniques. In general, four main groups were extracted, in this way recreating the ‘artistic palette’ of the unconventional photography of the artist. The four groups were the following: (1) Cr, Fe, K, potassium dichromate, and gum arabic bands characterized the dichromate salts; (2) Ag, Ba, Sr, Mn, Fe, S, Ba, gelatin, and albumen characterized the silver halide emulsions on the baryta layer; (3) the carbon prints were benchmarked by K, Cr, dichromate salts, and pigmented gelatin; and (4) the heterogeneous class of bromoil prints was characterized by Ba, Fe, Cr, Ca, K, Ag, Si, dichromate salts, and iron-based pigments. Some exceptions were found, such as the baryta layer being divided into gum bichromate groups or the use of albumen in silver particles suspended in gelatin, to underline the unconventional photography at the end of the 10th century.

## 1. Introduction

Pictorialism is an international style and esthetic movement that dominated photography during the later 19th and early 20th centuries, up to the 1930s in Italy, in which the photographer created an image rather than simply recording it. A photograph is like a painting, drawing, or engraving [1]. The photographic printing process was a handcrafted procedure rather than a standard procedure until it became an exclusively industrial product. Although the basic formulas and procedures to be followed were known (so much so that the same manufacturers produced the papers and sensitive materials suitable for the purpose), each operator had his own working method and his own secrets or tricks in making prints, jealously kept for himself and sometimes indicated on the back of the prints. The nascent photographic industry was countered by professionals’ refusal to use the photographic papers that were already prepared [2,3].

But alongside the internationally renowned photo amateurs, there have been many others, unknown or little-known, who have successfully dedicated themselves to these processes, producing images that are of considerable interest to researchers and photography scholars. Among them, Doctor Piero Vanni (Firenze 1884–Modena 1939) [4,5] represents an example of particular importance for the history of 20th-century photography in Friuli Venezia Giulia. The main contribution of this photographer–painter is the methodology followed and his aptitude for experimentation: he printed the same negative up to twenty or thirty times in the same format, varying formulas and developing new procedures.

Here, a selection of the photographic print collection of Piero Vanni depicting the *Valcellina* in the years 1911 and 1912 is analyzed. Thanks to his modus operandi, therefore, it is possible to detect the unconventional palette of an Italian regional photographic culture that is still unknown. Moreover, beyond this specific case, the evolution of social status and customs influenced the use, techniques, and materials of photography. In recent years, there has been a significant increase in interest in old photography, from conservatives, photographers, collectors, archives, and merchants to amateurs [6].

A widely used chemical process for black-and-white photography prints is the gelatin silver process [7,8,9]. Non-silver techniques are based on the photochemistry of other metallic salts (e.g., the toning of iron compounds) or the physical modification of certain binders exposed to light (e.g., the hardening of dichromate gum for pigment prints) [10]. The image layer is particularly difficult to examine by nondestructive analysis. Fourier transform infrared spectroscopy (reflectance mode) as X-ray fluorescence spectroscopy has already shown the potential to be a helpful tool in determining materials and processes for photographs. In pigment-based photographs, for instance, the identification of the organic components in the colloids is crucial for understanding the artistic processes [11]. Proteinaceous binder and inorganic salts have been identified in Italian photographs dated between 1890 and 1930 [12]. Spectroscopy combined with an unsupervised classification method was successfully used for the identification, classification, and authentication of prints and photography [6]. The current state-of-the-art method explores portable spectroscopy and multivariate analysis for the identification of paper and black colorants [6], to date gelatin silver prints [13], or to distinguish the type of binder [14]. The spectra are preprocessed for data analysis, such as normalization; averaged by a reducing factor of nine; put through spectral range selection; and exposed to other pretreatment methods because of the overlap and difficulty of assignment [14,15].

Here, a new methodology that combines nondestructive and noninvasive elemental and molecular analyses with unsupervised learning techniques was developed to extract fingerprints that are generally not easily identifiable because of the presence of the main binders (i.e., proteinaceous) from the photographic layers to unveil the stylistic practice and to define the ‘artistic palette’ of the unconventional photography around 1911–1912 pre-World War I.

## 2. Materials and Methods

### 2.1. Description of the Piero Vanni Collection

The ‘Piero Vanni’ collection is preserved at the Center for Research and Archiving of Photography (C.R.A.F.). A first macroscopic classification of the photographs as a function of the printing methods was carried out following Vanni’s annotations reported in the verso of the objects. Following this procedure, four main groups were identified: (1) gum bichromate, (2) silver particles suspended in gelatin, (3) carbon print, and (4) bromoil printing.

A total of thirty photographic prints were selected for this study, based on macroscopic observations, to collect spectroscopic data through a number of significant objects for each major category. The photographs are identified with an FFPV inventory code, which stands for the Piero Vanni Photographic Collection (for more details, see the Appendix A). The prints were classified into four photographs processes based on the printing technique, the coloring, type of paper, and handwritten notes on the back: gum bichromate, silver particles suspended in gelatin, carbon prints, and bromoil printing. Some of prints showed inconsistencies between the image layer and the handwritten notes. Due to degradation processes such as abrasions, stains, dust, and surface deposits, scaling, weakening of the margins, detachment from the primary support, yellowing of the binder, foxing, and fading, the prints required restoration. More details are provided in the Appendix A.

Despite using well-known techniques for that period, such as the gum bichromate process, and artisanal photographic processes, a preliminary macroscopic examination revealed inconsistencies with the author’s notes. As the author is known for creativity in the literature, a systematic mapping procedure was conducted for each image.

### 2.2. Measurements Strategy

The photographs were investigated by non-destructive and non-invasive spectroscopic techniques.

A total of 250 measurement points were carried out, with a minimum of 7 to a maximum of 10 measuring points for each photograph. The measurement points were strategically spaced to extract information on different areas, edges, centers, and areas of *chiaroscuro*, chosen on the *recto* (and one on the *verso* for each photograph) to map the image layer. This procedure will make it possible to identify the characteristic and proper elements of the image layer and the paper support, to discriminate between the two or more ‘’layers’’. For more details, see the Appendix A.

### 2.3. Spectroscopic Techniques

Energy dispersive X-ray fluorescence (ED-XRF) and Fourier transform infrared (r-FTIR in reflectance mode) spectroscopies were performed. A total of 250 spectra were collected for the ED-XRF and 88 spectra for the FTIR under the same experimental conditions.

(a)Energy dispersive X-ray fluorescence (ED-XRF). ED-XRF spectroscopy was performed by the XRAMAN spectrometer (XGLab S.R.L., Bruker Nano Analytics, Milano, Italy) [16]. It is characterized by a rhodium target X-ray tube operating at 50 kV and 200 mA, and the time of each acquisition it is fixed at 70 s. The detector is a large-area silicon drift X-ray detector with an active area of 25 mm^2^ and energy resolution of <135 eV measured on the MnKα line (5.890 eV) that allows element detection with atomic number Z > 11, with an input photon rate of up to 100.000 counts per second.(b)Fourier-transform infrared spectroscopy (FTIR). FTIR spectra were collected by a NICOLET iS5 spectrometer (Thermo Scientific, Waltham, MA, USA) equipped with a DTGS detector and KBr beam splitter. Spectra were sequentially recorded in the range of 4000–400 cm^−1^ with 128 scans and a resolution of 2 cm^−1^, yielding a total of 88 FTIR spectra (reflectance Mode–R%) and 4096 variables. Measurements were conducted with a Nicolet™ FTIR Thermo Scientific™ device equipped with a ConservatIR™ FTIR External Reflection Accessory for non-invasive and nondestructive analysis of large objects with a spot size of ~1.25 mm in diameter.

No normalization procedures were applied to the entire dataset (mathematical correction i.e., Kubelka–Munk transformations). Initially, all the spectra were visualized using the OMNIC and OriginPro® 2021 (OriginLab, 2023) software programs.

### 2.4. Optical Investigations

Optical investigations were also carried out for each print using a digital microscope (USB Microscope Jiusion) for surface observation with a maximum magnification of 1000× and a resolution of 1920 × 1082 p.

### 2.5. Machine Learning Analysis

Machine learning analysis were carried put through dedicated scripts developed via the software environment Colab notebook 6.5.1 (Jupyter Notebook service).

### 2.6. Algorithms Computed

The following techniques were applied to analyze the data.

Principal Component Analysis (PCA) (mean-centered)–Score and loading analysis: PCA provides a visual representation of the relationship between samples and variables, showing which variables influence the data samples or how they differ from each other. It is an unsupervised technique where features are assigned via a successive clustering process based on measures of the variance data’s distribution distance.

XRF raw data: The data matrix size is 13,997 × 23. The NIPALS algorithm used 100 as the value of iterations (the data are not noisy, and 100 iterations are sufficient to converge the data). As a validation method, a cross-validation set (the number of the sample is moderate) and a random set with twenty segments were chosen. A total of two optimal numbers of components explained a total variance for each PCA (total of variance cumulated: 96%, PC1 85%).

FTIR raw data: The data matrix size is 88 × 14,935. The Singular Value Decomposition (SVD) algorithm is used. As a validation method, a cross-validation set and a random set with five segments are chosen. A total of three optimal numbers of components explained a total variance for each PCA (total of variance cumulated: 93%, PC1 69%, PC2 21%, PC3 3%). Sample grouping: number of groups, three.

Kernel density estimation (2D): This is a technique for probability density function used to better study the probability distribution than using a traditional histogram method of smoothing parameterizations: a bivariate kernel density estimator on full bandwidth matrices; the number of points is 90 (all); the density is obtained with a first of 416 lowest density points; and the plot type is contour.

## 3. Discussion and Results

The results related to the four main categories are reported and described below.

### 3.1. Gum Bichromate

In the gum bichromate process, the potassium bichromate is added to gum arabic, which changes its water solubility when exposed to light (source: sunlight; UV is the active component) [17]. A pigment is mixed with the bichromate gum and applied to the paper. Afterward, it is washed and, once dry, it is placed under a negative and exposed to light. It is then washed with hot water with a brush, revealing the image.

Previous studies have mentioned recipes using gelatin or starch (chiefly arrowroot) and potassium dichromate, mixed with pigment [11,18]. The choice of pigment depends on the final visual effect the artist wants to create. 

XRF spectra (Figure 1) show the presence of Cr and K, key elements in the preparation based on the potassium dichromate, K_2_Cr_2_O_7_, in a greater quantity, acting as a photosensitizing agent, along with the use of a pigment as color (likely iron oxide), determining the reddish-brown coloring of some photographs in this group. The Ca contribution is associated with the paper substrate, and calcite or silicate can be from the pigment (iron oxide-based pigments) [19].

It is worth noting that barium (Ba) is the main pigment in one of the photographs in the gum bichromate group (FFPV_0063, Figure 1).

It is evident that Cr and Fe are elemental benchmarks, particularly in the first thirty points analyzed across four photographs (see Figure 2, right panel). These photographs are very similar in color and shape (details are provided in the Appendix A). Both Fe, at 6.40 keV, and Cr, 0 at 5.41 keV, show a consistent distribution across the photographs, suggesting an even spread of iron oxide pigment, except in the white areas. Cr is more abundant in the FFPV_0004 photograph, while Fe values are noted in the darker areas. Although the first four prints are similar, they exhibit slight differences in composition. In one instance, it seems that the gum bichromate was used on a layer of baryte, indicating a creative and experimental use of the photographic technique. The presence of Ba, as indicated in Figure 1, is only detected in FFPV_0063. Figure 3 demonstrates that the baryte layer was applied before the dichromate-pigment layer. The uniformity of the baryta treatment appears uniform, except for the top left corner (specifics are provided in the Appendix A). These results can be interpreted as evidence of Vanni’s method of application using a paint roller. This is supported by the results presented in Figure 2, where the edges show a sharp baryta layer.

**Figure 2 polymers-16-01850-f002:**
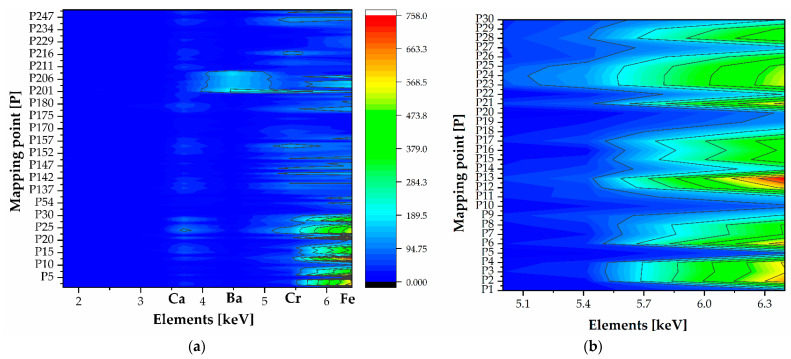
Two-dimensional density plot of the XRF gum bichromate print dataset. (**a**) Two-dimensional density plot in the range 1.74–6.40 keV to appreciate the Ca, Ba Cr, and Fe distribution. (**b**) Two-dimensional density plot in the range 4.93–6.40 keV to appreciate the Cr and Fe distribution.

**Figure 3 polymers-16-01850-f003:**
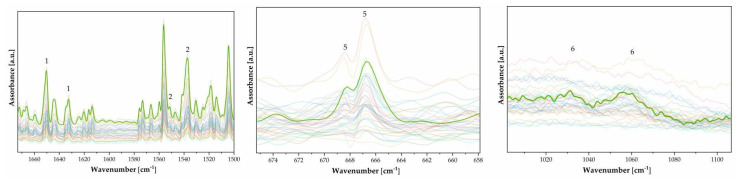
FTIR spectra (n. 40) of gum bichromate. Peak labeling refers to Table 1. The spectra are displayed with transparency to emphasize the overlapping peaks. One spectrum has been highlighted for improved figure readability.

**Table 1 polymers-16-01850-t001:** Main energy ranges and bond assignments: gum bichromate print.

n.	Wavenumber [cm^−1^]	Functional Group	Note
1	1650, 1633	νC–NH_2_, δN–H	Gelatin [19]
2	1537, 1550	νC–NH–C, δN–H	Gelatin [19]
3	990, 979 *	K_2_Cr_2_O_7_	Sensitizer [10]
4	223, 291, 409, 611	Fe_2_O_3_ ^1^	Pigment [10]
5	667, 668	SO_4_^2−^	Support [20]
6	1062, 1030	C–O, C–C	Cellulose chain [21]

* Weak. 1 Appreciable with micro FTIR.

### 3.2. Silver Particles Suspended in Gelatin

Silver gelatin is created by suspending light-sensitive silver halides in a gelatin binder, typically on a baryta paper support [17,21].

XRF spectra (n. 54) show a typical silver gelatin print processed on fiber-based paper. The light-sensitive emulsion comprises silver particles suspended in gelatin, which are then attached to a paper base coated with a layer of baryte (see Figure 4).

Here, iron is an optical impurity.

Figure 5 shows the functional groups of gelatin and their characteristic absorption frequency bands in Table 2. There are several regions such as the N-H stretching peak at 3300 cm^−1^ for the amide group overlapping with the O-H stretching peak at 3200 cm^−1^ for stretching H-bonds, which can be attributed to gelatin. Additionally, gelatin shows strong signals at 1650 and 1540 cm^−1^ (amide I and II peaks) typical for the C=O stretching amide group due to the proteinaceous image binder layer (Band Number 5, Table 2, Figure 2). The signal at 1526 cm^−1^ represents the N-H bending bond in gelatin. Intense absorbance in the region from 1240 to 1050 cm^−1^ and sharp features at 985 cm^−1^ are attributed to the multiple SO_4_^2−^ modes of barium sulfate. Figure 6a demonstrates the variation in the amount of metallic silver in the photographs, showing an increase in the concentration of Ag in one case (FFPV_0006_P45 and P46, see Figure 6). Moreover, in correspondence with these areas (violet and ochre lines), it is observed that the amide II band (1640, 1530 cm^−1^, Figure 6b) has a different absorbance: there is a reduced amount of albumen protein with an increased amount of metallic silver (FFPV_0006_P45, P46) compared to other photographs (FFPV_0029, P100, red star in Figure 6b), where the albumen is greater despite the amount of metallic silver.

### 3.3. Carbon Print

The carbon process relies on the light sensitivity of chromium present in dichromate salts suspended in pigmented gelatin, known as a dichromate colloid [17].

Photographs with the inventory numbers FFPV_0011 and FFPV_0019 belong to this group.

XRF spectra (n.18–Figure 7) show the presence of elements such as Ca, Cr, Fe, P, S, and K as elements. The presence of Ca is attributed to the paper substrate, while P is labeled as an impurity of the paper support. Carbon (C) is undetectable by the XRF device, due to its insensitivity to light elements below an atomic number of 11 (Na), so there is no carbon present in the XRF spectra. The presence of potassium (K) and chromium (Cr) indicates that the sensitizer used is potassium dichromate K_2_Cr_2_O_7_. Gelatin can be identified based on the presence of the FTIR spectral bands of amide I (located at 1626 cm^−1^) and amide II (1533 cm^−1^), which are typical of a wide range of protein materials. Additionally, bands typical of cellulosic materials can be identified. Figure 7 shows how the elemental spectra differ in quantity between the bright areas and shaded areas. As Cr increases, Ca and P increase, while S, K, and Fe remain almost unchanged.

Figure 8 shows the functional groups of gelatin along with their characteristic absorption frequency bands, which are listed in Table 3. Gum bichromate can be distinguished by the presence of FTIR spectral bands such as amide I, amide II, and amide III, which are typical of most protein materials, like gum arabic, used in paper preparation. It is also possible to identify bands typical of cellulosic materials.

### 3.4. Bromoil Printing

The bromoil process involves treating a silver gelatin print with a solution containing a dichromate salt. The gelatin hardens based on the amount of silver present. This process also takes advantage of the repulsion between oil and water, similar to lithography [17].

Photographs with the inventory numbers FFPV_0018, FFPV_0047, FFPV_0056, FFPV_p_0061, FFPV_p_0065, and FFPV_p_0067 belong to this group. These prints have diverse elemental and molecular composition. XRF spectra (n. 49, Figure 9), show the presence of Ba, Fe, Cr, S, Ca, K, Ag, and Si. The presence of Cr, along with pigments (iron-based), indicates the use of a dichromate salt used in the photographic process. Additionally, a baryta layer was also tried on four photographs (FFPV_0047, FFPV_0056, FFPV_p_0061, FFPV_p_0065). It is notable that most of the photographs’ pigment particles are visible, with the paper fiber being obscured (see Figure 9, optical observations), which is typical of the bromoil technique.

The presence of gelatin can be identified based on the FTIR bands in the reflection of amides (see Figure 10, Table 4). The distribution in the amide–protein fingerprint is not very concentrated, which suggests that the gelatin layer is not extensive (see Figure 10a).

### 3.5. Machine Learning

To uncover typical patterns and reveal hidden structures, a PCA was applied on the ED-XRF and r-FTIR datasets. A total of three optimal components for the system were obtained (for more statistical details please see the Section 2.2. subparagraph).

(a)PCA_Score_FTIR. In Figure 11, the score plot of the FTIR dataset (88 × 14,935 size) is displayed. This plot helps to assess the correlation between measurement points and interpret certain sample groupings, similarities, or differences in molecular spectroscopic data.

The distribution of measurement points does not align with the relevant photographic processes, reflecting Vanni’s innovative approach of reusing the same materials in different processes. Machine learning techniques are employed to detect minimal differences through extraction procedures [27,28]. Generally, all FTIR spectra exhibit similar characteristics, with significant vibrational peaks such as amides I, II, and III, which are typical of the Vanni collection’s photographs. Distinct groupings are observed, with the first pertaining to the verso side and concentrated in the third quadrant of the PCA, except for some points belonging to the recto. The latter are points where the emulsion/colloid layer is potentially thinner, leading to a greater contribution from the cellulosic chains of the paper support. Proximity between measurements points indicates a very small explained variance and significant similarity. For instance, the spectral response of photographic paper with the inventory number FFPV_0018 closely resembles that of paper with the inventory number FFPV_0064.

Another piece of evidence at the molecular level refers to the changes observed in the dichromate gums, as shown in the second quadrant of the PCA plot. Additionally, we can link PC2, which accounts for 21% of the total cumulative variance and describes one-fifth of the total dataset distribution, to the amount of albumen/gelatin present. This is indicated by the darker areas corresponding to zero to positive PC2 values (as shown by the arrow). These findings align with previous results (see Figure 6).

(b)PCA_Loading_FTIR. Figure 12 displays the loading plot of the FTIR dataset, which is 88 × 14,935 in size. The loading plot assesses the correlation of variables (spectroscopic benchmarks vibrations in specific spectral ranges) to interpret energy grouping, similarities, or differences in molecular spectroscopic data. Figure 12 shows the spectroscopic molecular benchmarks and the respective contributions of the individual PCs. The loading analysis in Figure 12 shows the correlation of the PCs and the characteristic spectroscopic regions or bands of the entire dataset. It also indicates the contribution of the cumulative information of the three components or those vibrational bands described by only one or more components. This allows for the identification of spectral regions and the contribution of all three components, such as the stretching vibration of the hydroxyl group (O–H) due to the water content or the typical bands of the proteinaceous binder or gum. Additionally, it is possible to extract esters (spectroscopic benchmarks at 1742 cm^−1^ [27]) highlighted, in part, by PC3, which accounts for 3% of the dataset.

**Figure 12 polymers-16-01850-f012:**
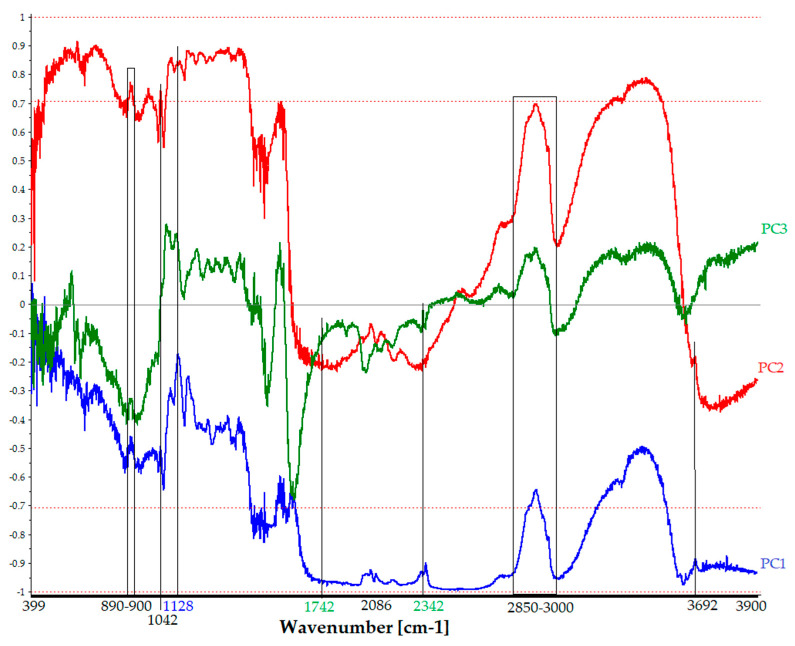
PCA correlation loading plot of r-FTIR spectra. Along the energy range, only some of the characteristic bands are shown, the variance contribution of each PC. The colors refer to the PCs to which they belong (blue, red, and green for PC1, PC2, and PC3, respectively). Details are given in Table 5.

**Table 5 polymers-16-01850-t005:** Main energy ranges and bond assignments: PCA loading (Figure 12).

Wavenumber [cm^−1^]	Functional Group	Note
3400–36,000	νO–H	Water content [23,29]–PC1, PC2, PC3
3692	νOH	Silicates [8]–PC1, PC2
2790	νCH_3_	Solvent [24]–PC3
2342	Me–O	Silica [30]–PC3
2357	νSi–O	Silicates [8]–PC1
2076	νMe–CO	Ag, Cu [31]–PC1, PC2, PC3
1650, 1633	νC–NH_2_, δN–H	Gelatin [19]
2850–3000, 1738–1757	νCH_2_	Gelatin [19]
1537, 1550	νC–NH–C, δN–H	Gelatin [19]
1437	νC–N–C	Gelatin [19]
890–900	β-glycosidic linkage	Cellulose/gum/starch ^1^ [32]–PC1, PC2, PC3
1042	ν–O, νC–C	Xylose [33]–PC1, PC2
1128	νC–Br	Methylene bromide [26]–PC1

^1^ Starch, for example, was an additive used to prepare matte papers as a papermaking component [34].

(c)PCA (Loading_XRF).

XRF analysis showed the presence of sulfur, calcium, iron, and barium, with the main contribution coming from barium. This suggests the presence of only barium sulphate. In addition to the main elements, Figure 13 also reveals interesting elements such as iodine and manganese, which were not initially observed in the spectra (due to being low-concentration elements where the corresponding lines could be very weak and difficult to distinguish from the background noise) [27,28]. The absence of platinum or gold was noted. The presence of Sr, Mn, and Fe is typical of the silver halide technology used in black-and-white photographs from the twentieth century and could be present in the emulsion layer [33]. Specifically, Sr and Cr are elemental benchmarks for baryte and baryte hardener, respectively.

The ED-XRF analysis investigated ‘’deeper layers’’, penetrating through the emulsion layer, paper, and reaching the cardboard beneath.

## 4. Conclusions

A new methodology that combines elemental and molecular analyses with unsupervised learning techniques was developed and tested on an unconventional Italian photographic print collection (the Piero Vanni collection, 1889–1939). This methodology aimed to uncover the artistic technique by extracting spectroscopic benchmarks. The results allowed for the distinction of hidden elements, such as iodine and manganese in the silver halide printing, or the highlighting of subtle differences in the same printing technique. Additionally, it revealed the stylistic practice. To interpret the machine learning results, a synergic integration of macroscopic observation, historical indications, and spectroscopic data was employed. This approach identified four main groups associated with the primary photographic techniques available in the first half of the 20th century. 

Dichromate salts with gum arabic print. Fourteen photographic prints have been characterized by sensitive dichromate salts and gum arabic. The prints were prepared using potassium dichromate as a photosensitizing agent, along with a pigment (likely iron oxide) responsible for the reddish-brown coloring in some of the photographs. An unconventional print (inventory number FFPV_0063) was found, as it used a baryta layer and an additional layer applied with a roller, as mentioned in the artist’s notes. A sharp-cut tool was used, particularly along the edges of the photograph FFPV_0063. Furthermore, FTIR analysis complements the XRF findings, identifying the presence of a sensitizer (potassium dichromate) bands, gum arabic, and potentially gelatin or starch. This group of prints, which is the largest numerically, exhibits a higher variance, as indicated by the spread of measurement points along the PCs.Silver halide emulsions on the baryta layer. A baryta layer on silver halide emulsions is identified in eight photographic prints. The spectroscopic elemental analysis revealed that the prints contain Ag, I, and Ba as elemental benchmarks, indicating the presence of colloidal and baryta layer parts. Additionally, Si was detected as part of the silicates added as additives as a matting agent. The PCA revealed the presence of an Il_α1_ peak at 3.94 keV, the second line K_β_ of a Ca peak at 4.01 keV, and a Mn peak at 5.90 keV (both hidden by the noisy signal—see Figure 4). The binder was identified as a mixture of gelatin and albumen and a N-H stretching band was detected, indicating the use of gelatin hardening during gluing. Machine learning techniques were employed to identify prints with a greater abundance of albumen and distinguish similar paper backgrounds used in the prints. Carbon prints. Carbon prints characterized two photographic prints using a dichromate colloid suspended in pigmented gelatin. XRF and FTIR analyses confirm the traditional process. In this instance, the artist emphasized the dark areas by incorporating the dichromate salt. The PCA’s score analysis revealed that this group of prints is the most consistent and homogeneous. The artist used commercially prepared papers, resulting in a stylistic uniformity among the prints.Bromoil prints. Bromoil prints are characterized by six photographic prints. This class of prints is diverse in its elemental and molecular composition. The photographic process involves using dichromate salt and pigments (iron-based), along with the presence of gelatin. The distinct vibrations of this type of photograph, particularly the oily part, were only revealed through PCA by the correlation loading.

Doctor Piero Vanni (1889–1939) was a meticulous experimenter: he printed the same negative several times using different techniques, making variations in formulas, recipes, and procedures. The methodology presented here allowed for the characterization of the photographs in terms of the materials used, the procedures of the photographic process, and also the recognition of elemental and spectroscopic benchmarks that were not visible from standard data analysis visualization.

The purpose of this work was to investigate the image layer using a methodology that leverages both the complementarity of non-invasive spectroscopic techniques and unsupervised techniques that allow the extraction of benchmarks (elemental and molecular) to define the unconventional techniques used by this author–painter. Being an example of artisanal pictorialism, even the most well-known printing processes could be diversified, as reported by some written testimonies [4,5]. This methodology can be applied in many fields, such as attribution studies [28].

## Figures and Tables

**Figure 1 polymers-16-01850-f001:**
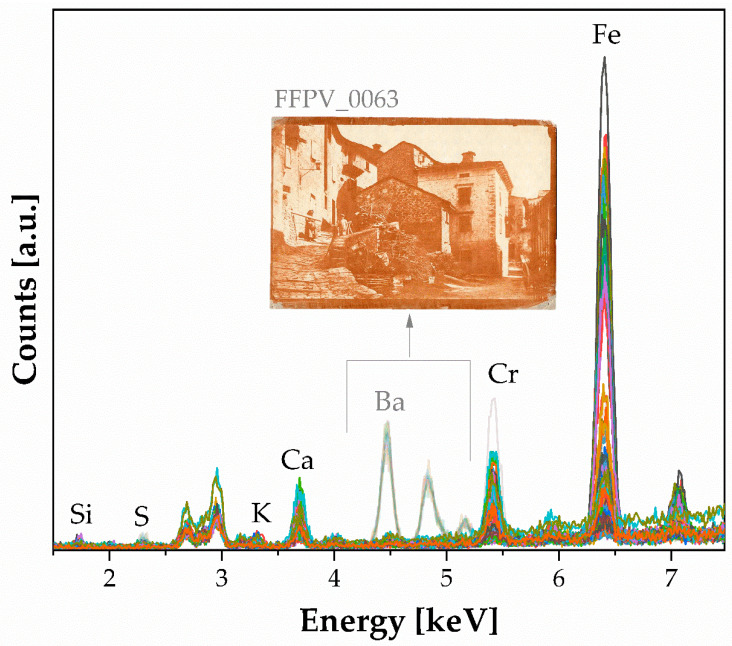
XRF spectra (n. 113) of gum bichromate. The photograph with the inventory number FFPV_0063 is a borderline case because it is the only sample showing the characteristic bands of baryte.

**Figure 4 polymers-16-01850-f004:**
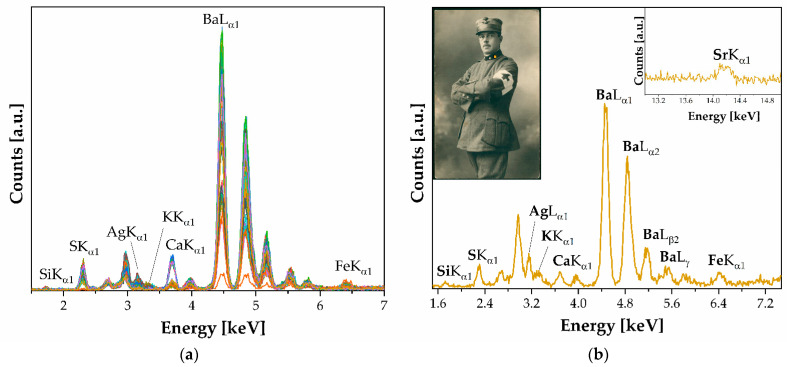
XRF of silver particles suspended in gelatin. Labeled peaks refer to the emission lines of K_α1_, except for Ba, whose emission line at 32.19 keV is not appreciable. (**a**) Silver salts gelatin spectra: the baryta layer (BaSO_4_) dominates in the spectra: all emission lines up to the L line are detected; (**b**) P36 spectrum.

**Figure 5 polymers-16-01850-f005:**
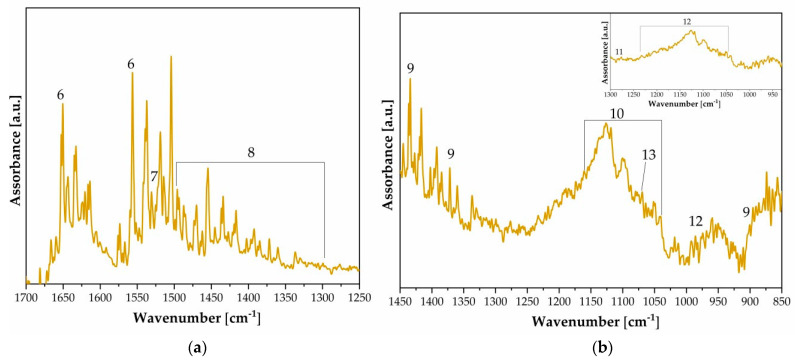
FTIR of silver particles suspended in gelatin: P36 spectrum. Labeled numbers refer to Table 2.

**Figure 6 polymers-16-01850-f006:**
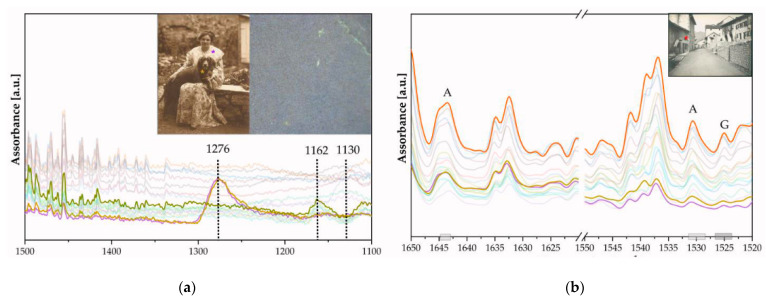
FTIR of silver particles suspended in gelatin. (**a**) The Ag signal is detected at 1276 cm^−1^ (FFPV_0006_P45, P46_supplementary materials), as highlighted by the optical investigations. Area investigated from the reverse side; cellulosic chain bonds dominate (1162, 1130 cm^−1^). The latter observation also gives some indication about the paper section and indirectly to the thickness of the various layers. (**b**) Selection of three spectra of silver particles suspended in gelatin (G) and albumen (A). The spectroscopic bands at 1640 cm^−1^ and 1530 cm^−1^ are attributed to amide II (albumen layer), νC=O, and δN-H in-plane bands, respectively.

**Figure 7 polymers-16-01850-f007:**
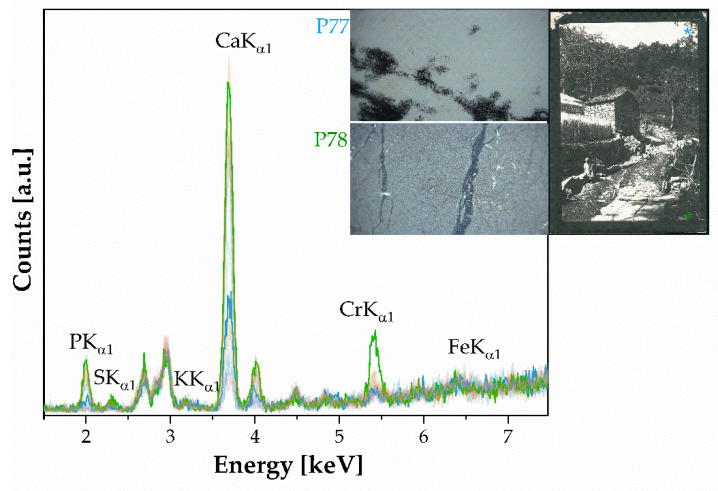
XRF of carbon particles suspended in gelatin. Labeled peaks refer to the emission lines of K_α1_. The insets referred to photograph with the inventory numbers FFPV_0019_P66, and P67. The optical investigations show how the appearance of the image layer changes significantly in the light areas compared to the dark areas: Cr is predominant in the dark areas.

**Figure 8 polymers-16-01850-f008:**
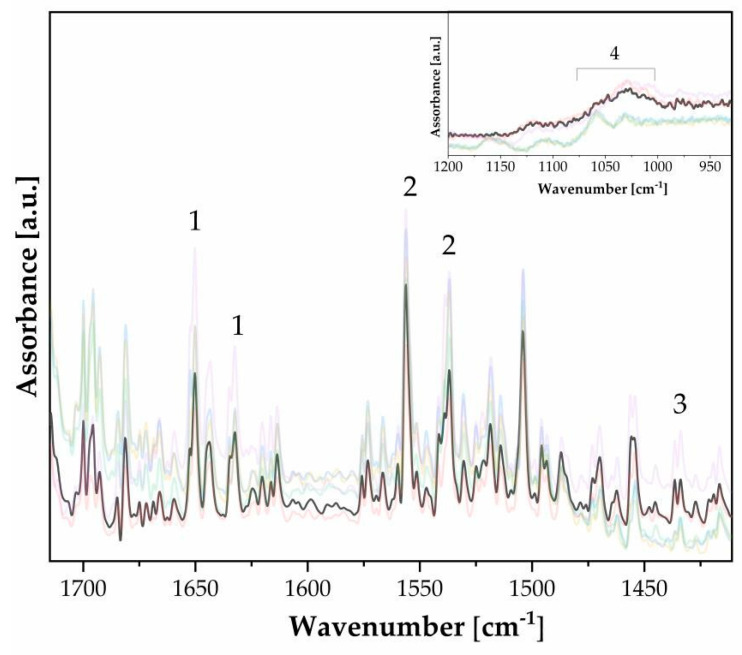
FTIR of carbon prints on gum bichromate.

**Figure 9 polymers-16-01850-f009:**
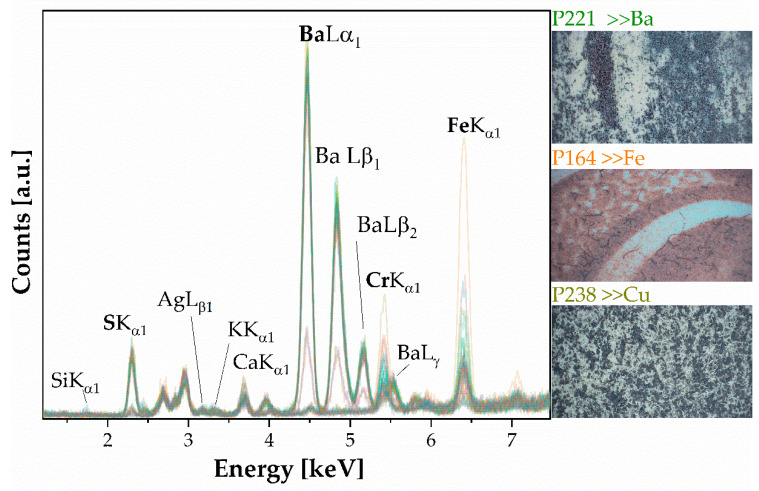
XRF of bromoil prints. Labeled peaks refer to the emission lines of K_α1_. The inset refers to photographs with the inventory numbers FFPV_0065_P221, FFPV_0056_P164, and FFPV_0067_P238. The optical investigations show how the appearance of the image layer changes significantly. Three photos have been selected, representing extreme case-studies according to the absolute abundance of the elements present: particularly rich in Ba, Fe, and Cu, respectively.

**Figure 10 polymers-16-01850-f010:**
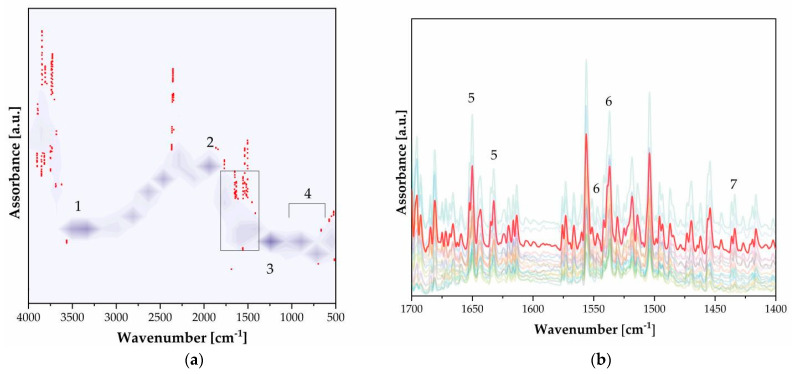
FTIR of bromoil prints. (**a**) Two-dimensional kernel density contour plot, where a photosensitive, photo-inhibited polymer is labeled with a solvent (Table 4). (**b**) Enlargement of the rectangle box where amides I, II, and III are labeled (Table 4).

**Figure 11 polymers-16-01850-f011:**
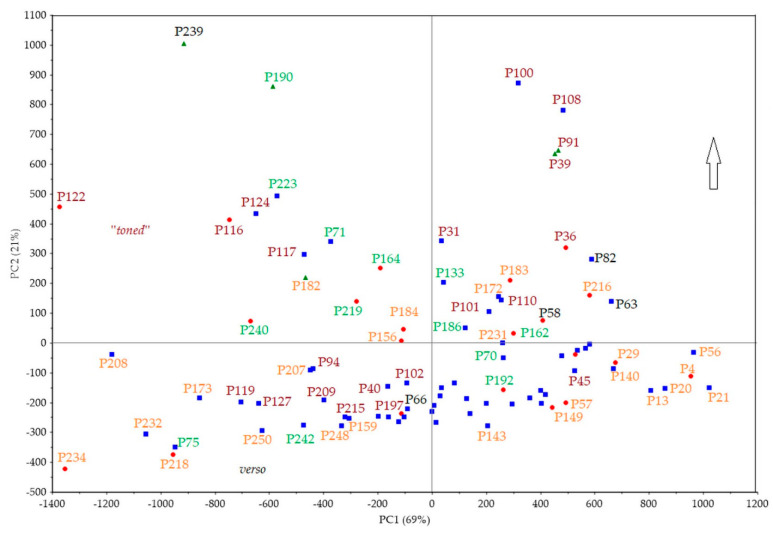
PCA, score plot of FTIR spectra. The measurement points are shown in four different colors according to the group they belong to (orange for gum bichromated, green for bromoil, dark burgundy red for silver gelatin, and black for carbon prints).

**Figure 13 polymers-16-01850-f013:**
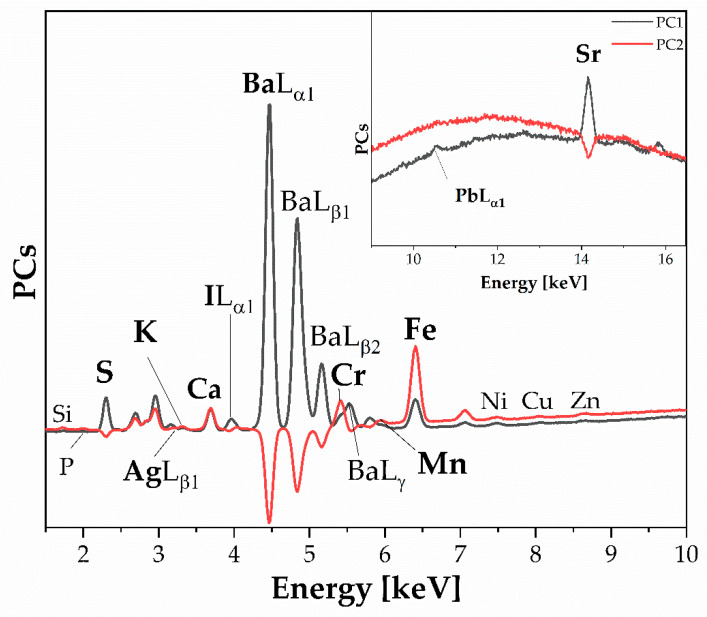
PCA, Loading plot. XRF raw data on 231 × 3997. The benchmark elements of the photographic techniques are reported in bold. The spectra obtained are computed by the ipynb file extension used by Jupyter Notebook.

**Table 2 polymers-16-01850-t002:** Main energy ranges and bond assignments: silver particles suspended in gelatin.

n.	Wavenumber [cm^−1^]	Functional Group	Note
1 *	3550–3200	νO–H	Alcohol/Phenol [21]
2 *	3500–3300	νN–H	Amine [21]
3 *	2950–2850	νC–H	Alkyl [21]
4 *	3100–3010	νC–H	Alkenyl [21]
5 *	3030	νC–H	Aromatic [21]
6	1650, 1540	νC=O	Amide I-II (protein layer) [7]
7	1526	δN–H	Gelatin/ammonium dichromate [21]
8	1500–1300	C=O, CONH_2_, NH	Gelatin/albumen/ammonium dichromate [12,21]
9	1430, 1375, 900	CH_2_, CH, C–O–C	Cellulose chain [8]
10	3340, 1100, 1062, 1030	C–O, C–C	Cellulose chain [8]
11	1276	Ag	Colloidal silver [8]
12	1240–1050, 985	SO_4_^2−^	Baryta layer [8]
13	1070	νSi–O ^1^	Silicate [8]

* Not shown. 1 Additive in the making of glossy paper [22].

**Table 3 polymers-16-01850-t003:** Main energy ranges and bonds assignments: carbon prints on gum bichromate.

n.	Wavenumber [cm^−1^]	Functional Group	Note
1	1650, 1633	νC–NH_2_, δN–H	Gelatin [19]
2	1537, 1550	νC–NH–C, δN–H	Gelatin [19]
3	1437	νC–N–C	Gelatin [19]
4	1062, 1030	C–O, C–C	Cellulose chain [7]
5	990, 979 *	K_2_Cr_2_O_7_	Sensitizer [11]

* Not appreciable.

**Table 4 polymers-16-01850-t004:** Main energy ranges and bonds assignments: bromoil prints from Density kernel plot (Figure 10).

n.	Wavenumber [cm^−1^]	Functional Group	Note
1	3334	–OH	Water [23]
2	1930	R–CHO	Hardening [24]
3	1243	CH_3_[Si(CH_3_)_2_O]nSi(CH_3_)_3_	Photosensitive poly(dimethylsiloxane) [25]
4	1000–900	νC–Br	Halite Emulsion [26]
5	1650, 1633	νC–NH_2_, δN–H	Gelatin [19]
6	1537, 1550	νC–NH–C, δN–H	Gelatin [19]
7	1437	νC–N–C	Gelatin [19]
8	1062, 1030 *	C–O, C–C	Cellulose chain [7]
9	990, 979 †	K_2_Cr_2_O_7_	Sensitizer [11]

* Not shown. † Not appreciable.

## Data Availability

The original contributions presented in the study are included in the article. The raw data supporting the conclusions of this article will be made available by the authors on request.

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
