# Peer review of "Spectroscopic Benchmarks by Machine Learning as Discriminant Analysis for Unconventional Italian Pictorialism Photography"

_polymers, 2024, doi:10.3390/polym16131850_

Round 1

Reviewer 1 Report

Comments and Suggestions for Authors

lines 92 - 107 shoud be a part of materials and methods as you have described the materials chosen for the analysis

lines 132 - 141: which FTIR method was used, KBr or ATR, please add

line 165: what does it stand "gum bicromate"? Are the photograps made by that process or? Please add better descripition. The same goes for other sections

line 167: which type of light, UV or else?

lines 166 -172: what propotion/ ratio of the used materials? Is it known?

line 180: such abbreviations should be previously described (FFPV_0063). The same goes for other abbreviations

Figure 1 shows wery large number of colored lines, what do they mean?

Table 1 shoul be written in uniqe form (functional group) somwhere is written bicromate, somwhere C-O. The same goes for other FTIR tables

Figure 3 is not readabe due to overlapping of multiple spectra. should be improvede. The same goes for other FTIR figures

Comments on the Quality of English Language

small grammar and style errors

Author Response

Comments and Suggestions for Authors (Reviewer 1)
We thank the reviewer for the comments and feedback on our manuscript. In the following, the answers point by point to the comments are reported.

Comment 1. lines 92 - 107 should be a part of materials and methods as you have described  the materials chosen for the analysis

Response comment 1
Lines 92 - 107 are moved in 'Materials and Methods' section and the text was revised accordingly.

Comment 2. lines 132 - 141: which FTIR method was used, KBr or ATR, please add.

Response comment 2
The device is a Nicolet™ FTIR Thermo Scientific™ equipped with a ConservatIR™ FTIR External Reflection Accessory. We add this information in the manuscript. 

Comment 3. line 165: what does it stand "gum bichromate"? Are the photographs made by that process or? Please add better description. The same goes for other sections.

Response comment 3
We confirm that the gum bichromate is one of the photographic processes used for these photographs. We provided a more detailed description in the revised sections (please see 188-198 lines). 

Comment 4. line 167: which type of light, UV or else?
Response comment 4 Sunlight is the source but only the UV is the active component. We revised the text for clarity  about the photographic process. 

Comment 5. lines 166 -172: what proportion/ ratio of the used materials? Is it known?

Response comment 5. 
As cited from reference (Vila, A.; S. A. Centeno; L. Barro; N. W. Kennedy. Understanding the gum dichromate process in pictorialism photographs: A literature review and technical study. Studies in conservation 2013, 176-188), printing process are generally described with indications about the proportions of the reagents, suggested exposure time and the hardening of the bichromate gum are reported. Exposure times depend on many variables: the amount of colloid, the type of coating, and all the unpredictability due to the artisanal 
nature of the process. Another variable is the distance from the UV lamp, which is more manageable or controllable. Therefore, the reagents proportion in the Vanni photographs are unknown because the procedure is entirely artisanal.

Comment 6. line 180: such abbreviations should be previously described (FFPV_0063). The same goes for other abbreviations. 

Response comment 6
The text now states the meaning of the inventory code. Please see lines 108-109.

Comment 7. Figure 1 shows very large number of colored lines, what do they mean?
Response comment 7. 
As described in the figure's caption, it represents 113 spectra from 113 measurement points appropriately chosen for the group of photographs carried out by gum bichromate process.

Comment 8. Table 1 should be written in unique form (functional group) somewhere is written bichromate, somewhere C-O. The same goes for other FTIR tables.
Response comment 8.
We revised the text accordingly. All tables are now consistent.

Comment 9. Figure 3 is not readable due to overlapping of multiple spectra. should be improved. The same goes for other FTIR figures

Response comment 9. 
We thank the reviewer for the comment. The number of spectra is very large, and it was chosen to make one of them stand out (from the others in transparency). The overlapping problem in this case must stand out because one of the aims of the work is precisely to highlight spectroscopic problems such as small differences. The added value of the ML algorithms is to discriminate small differences even when there is an overlapping problem of signals.
However, the resolution of all images was increased to improve the figures (1200 dpi) and most of the images have been redone.

With best regards,
Claudia Scatigno - on behalf of all the authors

Reviewer 2 Report

Comments and Suggestions for Authors

This manuscript provides a comprehensive study on analyzing unconventional Italian photographic prints from the Piero Vanni Collection using spectroscopic techniques and machine learning. The authors effectively combine energy dispersive X-ray fluorescence (ED-XRF) and Fourier transform infrared (FTIR) spectroscopy with unsupervised learning to extract and classify spectroscopic benchmarks. The study significantly contributes to cultural heritage research by introducing a novel, non-destructive methodology for analyzing historical photographic materials. With minor revisions, this manuscript is recommended for acceptance and publication in this journal.

Specific comments:

1.        While the study provides a thorough analysis of 30 selected photographs, a larger sample size could further validate the findings and ensure generalizability.

2.        The selection criteria for the photographs should be transparently discussed to avoid potential selection bias.

3.        The manuscript lacks details on the machine learning algorithms used, including parameter settings, training processes, and validation methods, making it hard to assess reproducibility and reliability.

4.        Including a comparative analysis with other non-destructive techniques using machine learning, such as Deep Transfer Learning and Generative Adversarial Networks (GANs), could provide a more comprehensive evaluation of the methodology's advantages and limitations.

5.        The study focuses on a limited number of photographic techniques. Including a broader range of techniques could provide more insights and demonstrate the methodology's versatility.

6.        The manuscript would benefit from a more explicit discussion on the study's limitations. Addressing potential sources of error or uncertainty in the spectroscopic and machine learning analyses would provide a more balanced perspective. Including specific suggestions for future research would enhance the manuscript's contribution to the field.

Author Response

Comments and Suggestions for Authors (Reviewer 2)
This manuscript provides a comprehensive study on analysing unconventional Italian photographic prints from the Piero Vanni Collection using spectroscopic techniques and machine learning. The authors effectively combine energy dispersive X-ray fluorescence (EDXRF) and Fourier transform infrared (FTIR) spectroscopy with unsupervised learning to extract and classify spectroscopic benchmarks. The study significantly contributes to cultural heritage research by introducing a novel, non-destructive methodology for analysing historical 
photographic materials. With minor revisions, this manuscript is recommended for acceptance and publication in this journal.

Response
We thank the reviewer for the comment. The feedback on the combination of ED-XRF and FTIR spectroscopy with unsupervised learning is greatly appreciated. We will carefully address the suggested revisions to further improve the manuscript. 

Specific comments:
Comment 1. While the study provides a thorough analysis of 30 selected photographs, a larger sample size could further validate the findings and ensure generalizability.
Response comment 1
The selected photographs are indeed a curated collection of artisanal, unconventional photographic processes by a single author. Although 30 photographs might seem limited, our mathematical computations have shown a 100% convergence, which is also confirmed by the total variance in the PCA results. Moreover, the dataset was meticulously chosen by experts in the field, focusing on instances where there was uncertainty (e.g., unclear or missing 
annotations on the back of the photos). 
Being an example of artisanal pictorialism, even the most well-known printing processes could be diversified, as reported by some written testimonies. Additionally, the number of spectra (number of measurement points) obtained, is 250 (just for the XRF dataset). The 30 images are therefore not homogeneous; otherwise, the variance would not have been achieved. 
It’s important to highlight that the study of the measurement points was carried out meticulously: measurement points were spaced to extract information on different areas, edges, centre, and areas of chiaroscuro, chosen on the recto, and on the verso for each photograph, to map the image layer as reported in 121-126 lines. This allowed for the detection of minimal differences even in the image layer of the same photograph. For this, in the future, this methodology can be applied to other cases of unconventional pictorialism, even with a similar number of objects as in this dataset or in many fields, such as attribution 
studies. Thank you again for your valuable comments.

Comment 2. The selection criteria for the photographs should be transparently discussed to avoid potential selection bias.
Response comment 2
The text was revised to highlight the selection criteria. 

Comment 3. The manuscript lacks details on the machine learning algorithms used, including parameter settings, training processes, and validation methods, making it hard to assess reproducibility and reliability.
Response comment 3
Thank you for constructive feedback. The validation and computation specifications werereported and described in the subsection 'Computed Algorithms'. We have nonetheless rewritten this part of the description for better understanding. We included all the necessary details regarding the machine learning algorithms used in our study, including parameter settings, training processes, and validation methods. We have specified additional information about the estimators in the case of the density plot. This ensures better 
understanding of the effectiveness and validity of our methodology, as well as enhance the reproducibility and reliability of our results. 

Comment 4. Including a comparative analysis with other non-destructive techniques using machine learning, such as Deep Transfer Learning and Generative Adversarial Networks (GANs), could provide a more comprehensive evaluation of the methodology's advantages and limitations.

Response comment 4. DL models such as CNN algorithms are widely used for the analysis of visual imagery, eliminating overfitting by Kernel function. CNNs are particularly suitable for data processing and data analysis, i.e., for object recognition, object detection, scene recognition, and semantic segmentation. Our data does not over-fit in the calibration stage, and we want to by-pass the preprocessing phase. Therefore, the use of certain algorithms (as RF) was 
completely unsuitable. Generative Adversarial Network (GAN) are applied to avoid the tradeoff for both high-phase sensitivity and high-resolution in specific applications. The GAN is a good option for image generation, but the latter is based on generator (G), and one discriminator (D). In our context, the discriminator is the output of our model: in PCA features are assigned via a successive clustering process based on measures of the distance of the 
variance data’s distribution. Machine learning unsupervised techniques such as principal component analysis are useful to distinguish between different techniques made with similar recipes using spectroscopic benchmarks as successfully demonstrated (Festa, G., Maggio, M.S., Teodonio, L. and Scatigno, C., 2023. Ancient handwriting attribution via spectroscopic benchmarks and machine learning: ‘Clavis Prophetarum’ by Antonio Viera. Expert Systems 
with Applications, 227, p.120328).

Comment 5. The study focuses on a limited number of photographic techniques. Including a broader range of techniques could provide more insights and demonstrate the methodology's versatility.
Response comment 5
The manuscript focuses on a new methodology that combines X-ray fluorescence and infrared analysis spectroscopy with unsupervised learning techniques - extracting spectroscopic benchmarks – applied to the study of the handmade photographic print by Piero Vanni Collection (1889 - 1939).
The printing processes presented here include the techniques most used in the late 19th and early 20th centuries. Although this is a pictorialism of a specific artist, the range of techniques covers the development methods of that era.
Thanks to the photographic extraction technique presented here, it is also possible to distinguish between techniques and the subtle differences within the same technique. In this sense, this study aims to demonstrate that this approach is successful and can be iterated to other cases.

Comment 6. The manuscript would benefit from a more explicit discussion on the study's limitations. Addressing potential sources of error or uncertainty in the spectroscopic and machine learning analyses would provide a more balanced perspective. Including specific suggestions for future research would enhance the manuscript's contribution to the field.
Response comment 6
We agree that discussing the study's limitations and addressing potential sources of error or uncertainty in the spectroscopic and machine learning analyses would provide a more balanced perspective. We hope that the answers to the points described above will clarify the type of approach used here and its purpose. Additionally, we will provide specific suggestions for future research to further enhance the impact of this research to the field. 

With best regards,
Claudia Scatigno - on behalf of all the authors

Round 2

Reviewer 2 Report

Comments and Suggestions for Authors

The authors have carefully revised this manuscript by addressing the reviewer' concerns. This reviewer would like to recommend accepting the revised manuscript.